# 3D Lung Tissue Models for Studies on SARS-CoV-2 Pathophysiology and Therapeutics

**DOI:** 10.3390/ijms231710071

**Published:** 2022-09-03

**Authors:** Roberto Plebani, Haiqing Bai, Longlong Si, Jing Li, Chunhe Zhang, Mario Romano

**Affiliations:** 1Center on Advanced Studies and Technology (CAST), Department of Medical, Oral and Biotechnological Sciences, “G. d’Annunzio” University of Chieti-Pescara, 66100 Chieti, Italy; 2Xellar Biosystems Inc., Cambridge, MA 02138, USA; 3CAS Key Laboratory of Quantitative Engineering Biology, Shenzhen Institute of Synthetic Biology, Shenzhen Institute of Advanced Technology, Chinese Academy of Sciences, Shenzhen 518055, China; 4University of Chinese Academy of Sciences, Beijing 100049, China

**Keywords:** SARS-CoV-2, 3D cultures, organ-on-a-chip, organoids, lung models, airways, alveolus, viruses

## Abstract

Severe acute respiratory syndrome coronavirus 2 (SARS-CoV-2), causing the coronavirus disease 2019 (COVID-19), has provoked more than six million deaths worldwide and continues to pose a major threat to global health. Enormous efforts have been made by researchers around the world to elucidate COVID-19 pathophysiology, design efficacious therapy and develop new vaccines to control the pandemic. To this end, experimental models are essential. While animal models and conventional cell cultures have been widely utilized during these research endeavors, they often do not adequately reflect the human responses to SARS-CoV-2 infection. Therefore, models that emulate with high fidelity the SARS-CoV-2 infection in human organs are needed for discovering new antiviral drugs and vaccines against COVID-19. Three-dimensional (3D) cell cultures, such as lung organoids and bioengineered organs-on-chips, are emerging as crucial tools for research on respiratory diseases. The lung airway, small airway and alveolus organ chips have been successfully used for studies on lung response to infection by various pathogens, including corona and influenza A viruses. In this review, we provide an overview of these new tools and their use in studies on COVID-19 pathogenesis and drug testing. We also discuss the limitations of the existing models and indicate some improvements for their use in research against COVID-19 as well as future emerging epidemics.

## 1. Introduction

Coronavirus disease 2019 (COVID-19), caused by the severe acute respiratory syndrome coronavirus 2 (SARS-CoV-2), is one of the most serious pandemics in human history. In addition to the respiratory system, this virus can impair the function of multiple organs, leading to death, particularly of older people with co-morbidities, such as cardiovascular diseases, chronic kidney and lung disease and diabetes [1]. Soon after the start of the pandemic, investigators from both academia and industry have been tirelessly seeking vaccines and treatments to combat COVID-19 [2,3,4,5,6,7,8]. However, the capability of the SARS-CoV-2 spike protein to mutate and therefore to increase viral transmission rate and escape immunosurveillance has raised additional challenges for the vaccination strategy. A key aspect in this battle against COVID-19 is the need for appropriate experimental models that reproduce human disease with high fidelity in order to generate data that can be rapidly translated to the clinic. In this respect, cell lines have been useful to elucidate mechanisms of viral invasion and infection [9], allowing the identification of the host receptor Angiotensin-converting enzyme 2 (ACE2), Neuropilin-1 (NRP1) and transmembrane serine proteinase 2 (TMPRSS2) as the main proteins involved in SARS-CoV-2 binding and entry into the host cells [10,11,12,13]. However, these 2D cultures revealed major limitations for antiviral drug testing. For example, while chloroquine (or its derivative hydroxychloroquine) showed an antiviral effect in the Vero E6 or Huh-7 cell lines [14,15,16], it failed to inhibit SARS-CoV-2 entry into human cells as well as demonstrate efficacy in clinical trials [17]. On the other hand, animal models, including primates, are not fully representative of human disease and often fail to predict the efficacy of antivirals in humans [14,18,19].

To fill this gap, 3D cultures that recapitulate human lung structure and physiology have been developed. In 2005, Fulcher et al. [20] reported that primary airway epithelial cells (HAE) can be grown on transwells under an air–liquid interface (ALI) and differentiated into a pseudostratified epithelium composed of the main airway cell types, i.e., basal, goblet, club and ciliated cells. Airway cells can be isolated from patients and used for studies of epithelial–stroma interaction when organized as organoids. These are self-organized 3D multicellular structures where the pseudostratified airway epithelium interacts with the extracellular matrix and maintains its main functions, i.e., mucus production and ciliary beating. Thus, these structures can reproduce in vitro the features of the human airway epithelium in vivo [21,22]. The respiratory epithelium organoid culture has been successfully used for research on cystic fibrosis (CF) and other pulmonary diseases [23,24], including viral infections [25,26,27]. However, a main limitation of the organoid models is the absence of vascularization and, therefore, the impossibility to study interactions between the viruses and cells of the immune system [28].

To overcome this limitation, more complex in vitro multicellular cultures have been developed. Today, the respiratory organ-on-a-chip technology represents the in vitro system with the highest structural and functional complexity. In addition to the 3D culture of lung cells under ALI, it includes vascular structures, allowing perfusion of drugs and immune cells under controlled shear, and it can be subjected to mechanical strain to mimic the respiratory cycle. This technology has been successfully applied to studies on COVID-19 [14,29], leading to significant advances in our knowledge of viral pneumonia and its treatment.

The main goal of the present work is to provide an updated review of the organoid and the organs-on-chip technologies, focusing on COVID-19 research. Challenges and improvements of these systems will be also discussed.

## 2. Lung 3D Cultures

The advent of porous membrane supports, such as transwells, has enabled the development of ALI cultures that reproduce in vitro with high fidelity the structure of the airway epithelium (Figure 1A). Indeed, all cell types found in the human airway in vivo, including ciliated, goblet, basal and club cells, as well as ionocytes can be observed in these cultures [20,21,22]. In addition, transwell cultures allow the investigation of the interactions among cell and tissue types, pathogens and the immune system [30,31,32]. These models have been extensively exploited to study CF, chronic obstructive pulmonary disease (COPD) and airway infections by bacterial and viral pathogens [33,34,35,36].

Compared to transwell cultures of airway epithelial cells, the culture of primary alveolar epithelial cells is less explored, partly because of limited cell availability. Primary alveolar type II cells (ATII) can only be cultured under 2D conditions for a limited period (3–7 days) before differentiation into alveolar type I (ATI)-like cells [37]. In addition, unlike airway epithelial cells, a consistent method to passage and expand primary ATII cells under 2D conditions has not yet been established. A few studies have investigated the culture of alveolar epithelial cell lines (such as A549 and NCI-H441) or inducible pluripotent stem cells (iPSC)-derived ATII cells under ALI, showing increased cell polarization, barrier formation, ion transport, and cell maturation [38,39,40]. However, none of these models supports both the maintenance of the ATII phenotype and the differentiation into ATI cells. Other efforts by coculturing alveolar epithelial cells with other cell types present in the lung alveolus, such as fibroblast and microvascular endothelial cells (MVEC), have obtained limited success [41,42,43]. Lung 3D cultures have also been used to study the immune response, monitored as cytokine production and lymph/monocyte, neutrophil migration [43,44].

Another 3D culture model that has attracted great interest in the field of lung biology and respiratory medicine is represented by lung organoids. These are 3D structures that can derive from pluripotent stem cells or adult stem cells undergoing spontaneous self-organization and differentiation in sphere structures with the apical side oriented toward the lumen or outward the medium [45,46,47] (Figure 1B). Organoids have been used to model the upper respiratory tract and even the distal alveoli in various lung diseases, including viral infections [48,49,50]. These 3D structures offer the advantage of allowing the study of the relationships between the epithelial tissue and the stromal component [51]. Furthermore, an organoid model that combines proximal and distal organoids, thus simulating in a certain way the crosstalk between the bronchi and the distal lungs has been recently developed [50,52]. Moreover, the inclusion of vascular networks by the incorporation of mesodermal progenitor cells has been recently described [53].

A further step towards the in vitro reconstitution of a human respiratory unit is represented by organ-on-chip technology. Ingber and his team pioneered the development of the lung-on-a-chip by growing lung epithelial cells and microvascular endothelial cells (MVEC) in two parallel microfluidic channels made of polydimethylsiloxane (PDMS), separated by a porous membrane [54,55]. Fluid shear stress and other mechanical forces, such as lung rhythmic breathing motions, can be introduced. Moreover, immune cells can be perfused through the vascular channel [29,56] (Figure 1C). This model has been optimized and utilized for studies on respiratory diseases, including CF, COPD, and lung infections [14,29,57,58]. Additional respiratory organ-on-chips with slightly different structures have been built by other groups. These are based on varying microfluidics platforms, mostly in PDMS, and composed of one or more channels or supports where to grow cells for differentiation into mature tissues [59,60,61,62]. These chips have been used for modeling viral infections and lung cancer or for studies on nanoparticle toxicity/transport in the lungs [59,60,61,62].

## 3. Three-Dimensional Systems in COVID-19 Research

Since the beginning of the COVID-19 pandemic, 3D in vitro cultures including airway organoids, static 3D co-cultures and organs-on-chip have been used for mechanistic studies on SARS-CoV-2 infection, as well as for drug repurposing, and testing of novel antiviral therapeutics. Compared to animal models, these systems demonstrated a higher fidelity to human disease and a high prediction rate of drug efficacy [14,63].

### 3.1. Lung Organoids

Organoids have been a reference model during the fight against COVID-19. It has been reported that both airway and alveolar organoids can be successfully infected with influenza and SARS-CoV-2 viruses [64,65]. Primary cell-based organoids provided a useful system for understanding SARS-CoV-2 pathogenesis and identifying effective antiviral drugs. Indeed, several SARS-CoV-2 inhibitors have been identified using the organoid model. Han et al. reported the efficacy of imatinib, mycophenolic acid and quinacrine dihydrochloride, against SARS-CoV-2 in a human physiologically relevant setting [66]. The in vivo antiviral efficacy of these drugs has been explored and confirmed, leading to the conclusion that hPSC-organoids can be useful to study SARS-CoV-2 infection and provide a valuable model for the identification of anti-COVID-19 molecules. This work and other studies have led to clinical trials to evaluate the efficacy of imatinib against COVID-19 [67]. Another study by Sano et al. [68] describes the anti-SARS-CoV-2 efficacy of camostat, remdesivir and EIDD-2801 in bronchial organoids. The organoid models also allowed the study of airway regeneration after SARS-CoV-2 infection by monitoring the replacement of ciliated cells destroyed by the virus with new ciliated cells differentiated from basal cells [68]. Of note, remdesivir is currently administered to patients with COVID-19. Spitalieri et al. [69] developed 3D complex lung organoid structures (hLORGs) starting from human iPSCs. Using the hiPSC-derived hLORGs, they identified two immunotherapeutic candidates for COVID-19 treatment, a tetravalent neutralizing antibody (15033-7) targeting the spike protein, and a synthetic peptide homologous to the dipeptidyl peptidase-4 (DPP4) receptor on host cells. Both Ab15033-7 and DPP4 significantly inhibited infection by SARS-CoV-2 S pseudovirus in hLORGs. Although the efficacy of these molecules against native SARS-CoV-2 needs further exploration, this study demonstrates that hiPSC-derived hLORGs could provide an alternative system for testing therapeutics against COVID-19. Tindle et al. [50] created adult lung organoids (ALOs) that are composed of both proximal airway and distal alveolar epithelium for modeling the SARS-CoV-2 infection and associated host immune responses. They demonstrated that the ALO model of the SARS-CoV-2 infection better recapitulates the transcriptomic signatures of respiratory samples from diverse cohorts of COVID-19 patients, compared with other existing SARS-CoV-2-infected lung models, due to the fact that this model contains both proximal and distal alveolar signatures. In this respect, this model could serve as a preclinical screening system for the identification of drugs that target both local and immune responses. Along these lines, Lamers et al. [49] described a human 2D ALI culture system wherein alveolar, basal and rare neuroendocrine cells are derived from 3D self-renewing fetal lung bud tip organoids. They showed that SARS-CoV-2 can readily infect these cultures mainly targeting surfactant protein C-positive alveolar type II-like cells and that it can be inhibited by a low dose of interferon lambda 1. Distal lungs were also successfully infected by SARS-CoV-2, indicating that this model can be useful to study alveolar pathogenetic processes. On the other hand, Salahudeen et al. [48] developed a long-term feeder-free culture system of human distal lung organoids derived from single adult human alveolar epithelial type II (AT2) or KRT5^+^ basal cells. They demonstrated that the organoids enabled the analysis of SARS-CoV-2 infection of the distal lung and revealed that SCGB1A1^+^ club cells are targeted by SARS-CoV-2.

### 3.2. Three-Dimensional Cultures on Transwells

ALI cultures provide a more physiologically relevant environment compared to conventional 2D cell cultures. The main advantage of epithelial ALI cultures is the possibility to deliver viral particles through the airway route, thus allowing viral replication and transmission at the apical sides of the lung epithelium. Thanks to these research tools, the main mechanisms of SARS-CoV-2 viral entry, which involve the binding of the receptor binding domain (RBD) of the SARS-CoV-2 spike protein to ACE2, were identified soon after the onset of the pandemic. Zhang et al. developed recombinant human ACE2-Fc fusion protein (hACE2-Fc) and constructed an ALI model, consisting of lung epithelial cells, obtained by fiberoptic bronchoscopy and brushing of the airway walls, grown in ALI using transwells [70]. These investigators demonstrated that hACE2-Fc potently neutralized the SARS-CoV-2 virus as efficiently as neutralizing antibodies. Along these lines, Djidrovski et al. [71] obtained basal-like cells by differentiating iPSCs and used these cells to generate airway epithelial equivalents by ALI culture. They seeded the basal airway-like cells onto the apical side of 24-well plate cell culture inserts with a transparent membrane and fed them from the basal chamber to induce differentiation. The differentiated cell types included functional ciliated cells, capable of secreting mucus, which were readily infected by SARS-CoV-2. The infected cells secreted cytokines at levels comparable to those detected in the airway epithelium in vivo following SARS-CoV-2 infection. In another study, two models were constructed: ALI cultures of proximal airway epithelium and alveolarsphere of distal lung AT2 cells [52]. The distal lung epithelial cells were mixed with human lung fibroblast cells and resuspended in Matrigel. Both models were susceptible to SARS-CoV-2 infection, resulting in an autonomous pro-inflammatory response. Remdesivir strongly inhibited SARS-CoV-2 infection and/or replication in both models. However, a limitation of these constructs is the lack of immune and endothelial components. Notably, SARS-CoV-2 clinical isolates infected a 3D human respiratory epithelial cell model, which was developed using primary respiratory epithelial cells differentiated on transwells at ALI, with the highest expression of SARS-CoV-2 viral RNA at 24 h post infection. Interestingly, two long noncoding RNAs, LAS1 and TOSL were upregulated in both nasal swabs from COVID-19 patients and this 3D culture model [72], suggesting that these innate immune modulators may play a role in SARS-CoV-2-induced innate airway mucosal responses.

### 3.3. The Organ-on-a-Chip Technology

Dynamic models such as Lung-on-a-Chip have several advantages over static 3D models. First, some of them allow the application of mechanical forces that simulate alveolar contraction and expansion, which is crucial for the activation of innate immunity [29]. Furthermore, the possibility of perfusing leukocytes through the vascular channel allows the analysis of the anti-viral immune response [73]. A recent work used this model to monitor influenza virus evolution during multiple serial passages, identifying relevant clinical mutations and allowing the resistance to antiviral drugs to be studied [74]. Si et al. exploited the Airway-on-a-Chip, built with human-differentiated ciliated bronchial epithelium and lined with microvascular endothelial, to study SARS-COV-2 and the influenza virus infection as well as the neutrophilic responses to viral infection [14]. The use of Airway-on-a-Chip also demonstrated the ineffectiveness of hydroxychloroquine and chloroquine, contrary to what was seen in vitro [75], but consistent with the observations in non-human primates [76] and data from clinical trials [77,78,79]. In addition, Si et al. [14] revealed that clinically relevant doses of amodiaquine inhibited infection of human airway chips by pseudotyped SARS-CoV-2. The differences between amodiaquine and the other related antimalarial drugs, hydroxychloroquine and chloroquine, could be partially explained by their effects on the proteome of airway epithelial cells. In fact, quantitative proteomics analysis revealed that amodiaquine has different and broader effects on the host proteome, especially in relation to the proteins involved in the regulation of ciliary functions, compared with hydroxychloroquine and chloroquine. The in vivo prophylactic and therapeutic efficacies of amodiaquine were validated in hamsters challenged with native SARS-CoV-2. Artesunate-amodiaquine is being tested in a clinical trial for the treatment of COVID-19 in Africa [80].

The organ-on-chip has been used by several groups to study the response to SARS-CoV-2 infection in the distal lungs [29,81,82]. Epithelial cells are not the only cell types attacked by SARS-CoV-2. Indeed, microvascular thrombi and endotheliitis were found in lungs of patients who had died from COVID-19 [83,84,85,86]. Moreover, lung microvascular endothelial cells (LMVEC) express high levels of NRP1 [10], an alternative receptor for SARS-CoV-2 entry [13]. Thacker et al. [81] used the Lung-on-a-Chip model to study SARS-CoV-2 infection and its effects on endothelial cells (EC). They observed damage to the endothelial barrier integrity, confirming the pro-coagulant effects of SARS-CoV-2 observed in animal models and autoptic evidence from human subjects. Of note, ACE2 expression increased by ~10-fold in LMVEC on-chip co-culture conditions under shear stress, highlighting the relevance of a physiological microenvironment to studying viral infection, thus confirming on chip the alveolar-capillary injury during SARS-CoV-2 infection observed in a co-culture transwell model [30]. Finally, Bai et al. [29] examined tissue-level host responses of a human alveolus chip, including barrier injury, ATII cell death, tissue regeneration, and recruitment of B and T lymphocytes and monocytes, following infection with respiratory viruses such as H3N2, H5N1 and coronaviruses. This study revealed a critical role of mechanical forces in shaping lung innate immunity and identified the receptor for the glycation end product (RAGE) pathway as a major drive during mechanotransduction, in addition to being an amplifier of aberrant host responses in viral pneumonia. The antiviral properties of the RAGE inhibitor azeliragon and its synergist effect with molnupiravir were also uncovered by this study. A summary of the main findings obtained with these models is reported in Table 1.

## 4. Discussion

In this review, we summarize the current knowledge on the use of respiratory 3D cultures, namely, organoids and organs-on-chips, for studies on the pathophysiology and pharmacology of viral infections. This field of investigation has become dramatically relevant with the outburst of the COVID-19 pandemic that has so far affected more than half a billion people causing almost six and half million deaths. A significant number of studies have consistently demonstrated that organoids and organs-on-chip constitute a more reliable system to model human respiratory viral infections, better than traditional 2D cell cultures. These 3D lung tissue models, whose characteristics are illustrated in Table 2, faithfully recapitulate human pathophysiology and host responses to the SARS-CoV-2 infection, each providing valuable information for the development of better treatment options against the COVID-19 pandemic, especially when the limitations of the available animal models are evident.

Indeed, mice, ferrets, hamsters and macaques [87,88,89] have been used as preclinical models of SARS-Co-2 infection; however, these models and even non-human primates often do not fully recapitulate the human disease, due to species-specific differences in receptors distribution, protease expression and host immune responses [18,19]. Thus, 3D cultures have been instrumental for the study of interactions among pathogens, tissues and the immune system, representing a significant step forward in in vitro studies of respiratory infections. Data from different groups, using different models have consistently confirmed that the more complex 3D cultures, namely, organoid and organ chips, have greatly expanded our toolbox to study virology in vitro and helped understand processes, such as viral entry, replication, and host innate and adaptive immunity, thus enabling the identification of more effective therapeutic targets and drugs [14,29]. In the specific case of COVID-19, the Lung and Airway-on-a-chip has provided a reliable platform to study how SARS-CoV-2 infects human cells and triggers the complex host-virus interplay and the immune system [14,29].

Studies with this model have proven clinical relevance. In fact, amodiaquine, which emerged as an effective anti-SARS-CoV-2 agent in the Lung-on-a-chip [14], has been approved for clinical trial [80], likewise, remdesivir [68], which is currently used to treat patients with COVID-19. Thus, the high predictive power of drug efficacy represents a strong indication for the use of the Airway and Lung-on-a-chip in preclinical studies.

However, these chips are not the only ones used for research on viral diseases. Given that COVID-19, as well as other coronaviruses, can affect several organs, including kidneys, and the gastrointestinal, cardiovascular and nervous systems [90,91,92,93], 3D models of other organs have been used. Bein et al. [94] recently modeled an immunocompetent human Intestine-on-a-Chip to study enteric NL63 coronavirus infection and treatment, whereas Helms et al. [95], using kidney organoids derived from pluripotent stem cells, showed that SARS-CoV-2 can directly infect and damage kidney tubular epithelial cells. They also demonstrated that kidney organoids can be used for drug testing.

In conclusion, even though a definitive cure for SARS-CoV-2 has not yet been discovered, 3D tissue models, such as lung organoids and lung-on-a-chip, have been extraordinarily useful in combating the COVID-19 pandemic by providing a vast amount of data on key clinical and therapeutic aspects of this disease. However, there are some limitations of this technology that need to be considered. Working with advanced cultures such as organoids or organ-chips requires extreme accuracy, particularly during the phase of cell differentiation and tissue assembly. Therefore, to obtain reliable results, an adequate number of replicates should be planned. On the other hand, primary cells are not always available, and, therefore, particularly in the perspective of personalized medicine, the use of iPSC-derived cells should be considered. Another limitation, particularly in the case of organ-on-chip technology, is the low yield of materials for conventional biological assays, such as Western blot or flow cytometry. This, however, can be circumvented by combining multiple chips or using high-throughput assay technologies, such as total RNA or single-cell sequencing [96,97].

## 5. Future Directions

Despite the indisputable utility of the current 3D models, the possibility of building more complex systems that include all tissue and immune components of an airway functional unit and to interconnect this unit with chips of other organs, affected by the same disease, remains a fascinating perspective. For instance, the current 3D technologies related to the respiratory system could be implemented by including fibroblasts and alveolar macrophages to study the inflammatory immune response. Moreover, the 3D culture can be maintained only for a few weeks. Thus, extending the culture duration will allow the study of chronic processes associated with viral infections, for example, lung fibrosis or the role of memory B- or T-cell responses, a crucial aspect in the field of virology [98].

## Figures and Tables

**Figure 1 ijms-23-10071-f001:**
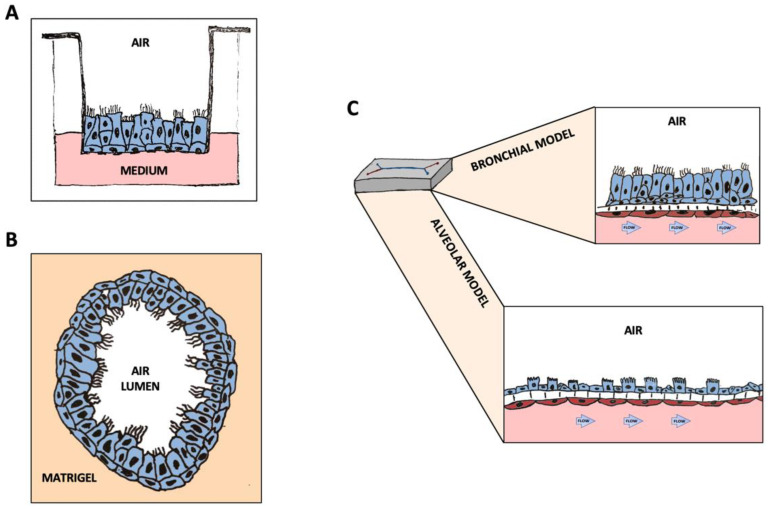
Graphic representation of the 3D structures discussed in this manuscript: (**A**) Airway Epithelial cells in ALI culture on transwells. (**B**) Airway organoids composed of all cell types, differentiated and oriented towards the inner lumen. (**C**) Differentiated lung-airway (top) and lung-alveolus (bottom) chips. The differentiated Airway-on-a-Chip is composed of ciliate, basal, goblet and club cells in the top channel, interfaced with pulmonary microvascular endothelial cells perfused using a microfluidic device. The differentiated alveolus chip is composed of ~45% type II and ~55% type I lung alveolar cells, forming together an in vitro surfactant-producing human alveolar structure.

**Table 1 ijms-23-10071-t001:** Summary of 2D/3D in vitro cultures in COVID-19 research.

Technology	Major Findings	Culture Type	Reference	Date
Conventional cell cultures	Identification of ACE2 as a binding receptor for SARS-CoV-2.	2D cultures	[11]	2020
	SARS-CoV-2 receptor ACE2 and TMPRSS2 are primarily expressed in bronchial transient secretory cells.	2D cultures	[12]	2020
	Neuropilin-1 is a host factor for SARS-CoV-2 infection.	2D cultures	[13]	2020
	Neuropilin-1 facilitates SARS-CoV-2 cell entry and infectivity.	2D cultures	[10]	2020
Transwells	Confirmation of the endothelial damage and increased epithelial and endothelial inflammatory status related to SARS-CoV-2 infection.	Co-culture on transwell	[30]	2020
	Identification of increased levels of LAS1 and TOSL lnc RNAs in both nasal swabs from COVID19 patients and 3D cultures.	3D bronchial epithelium on transwell	[72]	2021
	Proof of SARS-CoV-2 infection of surfactant protein C-positive alveolar type II-like cells and efficacy of interferon lambda 1.	2D ALI culture system alveolar cells	[49]	2021
	Development of recombinant human ACE2-Fc fusion protein using cells obtained by brushing of the airway walls.	3D ALI culture	[59]	2021
Organoids	Description of anti-SARS-CoV-2 efficacy of camostat, remdesivir, and EIDD-2801.	3D airway organoids	[68]	2022
	Description of imatinib, mycophenolic acid and quinacrine dihydrochloride efficacy against SARS-CoV-2.	Distal lung organoids	[66]	2020
	Description of SCGB1A1^+^ club cells as targets for SARS-CoV-2.	Distal lung organoids	[48]	2020
	Identification of a tetravalent neutralizing antibody targeting SARS-CoV-2 spike protein and of a synthetic peptide homologous to dipeptidyl peptidase-4 receptor on host cells as candidates for COVID-19 treatment.	Distal Lung-organoids from iPSCs	[69]	2022
	A combined model susceptible to SARS-CoV-2 infection.	3D airway + 3D aleolarspheres	[52]	2020
	Development of lung organoids composed of both proximal airway and distal alveolar epithelium for SARS-CoV-2 infection.	Complete lung organoid	[50]	2021
Organ-on-a-chip	Drug efficacy against SARS-COV-2 and influenza viruses and study of neutrophil responses to the viral infection. Proof of hydroxychloroquine and chloroquine inefficacy and amodiaquine efficacy.	Lung-airway-chip	[14]	2021
	Monitoring influenza virus infection during multiple serial passages, identifying relevant clinical mutations and resistance to antiviral drugs.	Lung-airway-chip	[74]	2021
	Description of Endotheliitis in the lung-on-a-chip model after SARS-CoV-2 infection.	Lung-alveolus-chip	[81]	2021
	Analysis of the innate response to H3N2, H5N1 and coronaviruses during breathing and identification of azeliragon and molnupiravir as potential antiviral drugs.	Lung-alveolus-chip	[29]	2022

**Table 2 ijms-23-10071-t002:** Comparison of the available in vitro lung models to study SARS-CoV-2 infection.

Model Type	Ease of Culture	Cost	Possibility of Coculture	Duration of Culture	Throughput	In-Vivo Mimicry	Predictive Power of Pharmacological Responses
2D culture	Easy	Low	No	Short	High	Low	Low
Transwell culture	Easy/Medium	Medium	Yes	Medium/Long	Medium/High	Medium	Medium
3D Lung Organoids	Medium/High	High	No	Medium/Long	High	Medium	Medium
3D Organ Chips	High	High	Yes	Medium/Long	Low	High	High

## Data Availability

Not applicable.

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
