# Peer review of "3D Lung Tissue Models for Studies on SARS-CoV-2 Pathophysiology and Therapeutics"

_ijms, 2022, doi:10.3390/ijms231710071_

Round 1
Reviewer 1 Report
Plebani et al. have submitted an interesting review paper on lung culture models that might be used in COVID-19 research. The paper is interesting to read, thorough, and fits the scope of the Journal and the SI. There are, however, some minor improvements that can be made:
- Table 1 should be restructured according to the journal template, and perhaps several sections can be introduced to separate the different models into categories within the table
- all citations, including clinical trials, should be formatted according to the instructions for authors
- the introduction of a new table in the Discussion section, to present the advantages and disadvantages of the various presented models would be very beneficial to the readers
- titles of sections 2 and 3 should be renamed to improve clarity and distinction; in the current form, they both address 3D structures, organoids, and so on; please find a clearer way to separate the content, otherwise, I recommend to fuse the two chapters and introduce subsections accordingly.
- a thorough English check should be performed - several grammar and spelling errors throughout the text (e.g. caption of Figure 1).
Respectfully submitted,
Author Response
We thank this reviewer for the time and effort spent in revising our work. Below our response to the criticisms raised by this referee.
Point 1: Table 1 should be restructured according to the journal template, and perhaps several sections can be introduced to separate the different models into categories within the table
Response 1: Thank you for your suggestion. We have updated Table 1 according to the journal template and grouped the publications by technology. An additional column “Publication Date” has also been added to the table.
Point 2: All citations, including clinical trials, should be formatted according to the instructions for authors
Response 2: In the revised version, we have corrected all citations, including clinical trials, according to the instructions for authors.
Point 3: The introduction of a new table in the Discussion section, to present the advantages and disadvantages of the various presented models would be very beneficial to the readers
Response 3: Thank you for this suggestion. We have introduced a new table in the updated version of the manuscript.
Point 4: Titles of sections 2 and 3 should be renamed to improve clarity and distinction; in the current form, they both address 3D structures, organoids, and so on; please find a clearer way to separate the content, otherwise, I recommend to fuse the two chapters and introduce subsections accordingly.
Response 4: Thank you for your comment. We have reorganized the two chapters and introduced subsections based on your indication.
Point 5: A thorough English check should be performed - several grammar and spelling errors throughout the text (e.g. caption of Figure 1).
Response 5: Thank you for your comment, we carefully checked for English and corrected several grammar and spelling mistakes, hoping that we did not miss any.
Reviewer 2 Report
Overall, this article provides comprehensive review about the use of 3D lung tissue models for studies on SARS-CoV-2 patho-physiology and therapeutics. The manuscript is well-written. Thus, I just have minor suggestions.
1. Please add a briefly summary about the clinical implication.
2. May add some discussion in addition to lung because SARS-CoV-2 can also involve extrapulmonary systems.
2. Please check the references format.
Author Response
We thank this reviewer for the accurate reading of our work. We updated the manuscript in order to address all points raised by this referee.
Point 1: Please add a briefly summary about the clinical implication.
Response 1: Thank you for this suggestion. In the discussion, we have included a paragraph that highlights the clinical implication of the use of the 3D models.
Point 2: May add some discussion in addition to lung because SARS-CoV-2 can also involve extrapulmonary systems.
Response 2: Thank you for your comment. In the revised discussion, we have introduced a paragraph on extrapulmonary 3D models for SARS-CoV-2 research.
Point 3: Please check the references format.
Response 3: We have corrected the references format.